# Sleep Apnea and Amyotrophic Lateral Sclerosis: Cause, Correlation, Any Relation?

**DOI:** 10.3390/brainsci14100978

**Published:** 2024-09-27

**Authors:** P. Hande Ozdinler

**Affiliations:** Department of Neurology, Feinberg School of Medicine, Northwestern University, Chicago, IL 60611, USA; ozdinler@northwestern.edu

**Keywords:** CSF, glymphatic system, neurodegeneration, spine loss

## Abstract

Amyotrophic lateral sclerosis (ALS) is a motor neuron disease with progressive neurodegeneration, affecting both the cortical and the spinal component of the motor neuron circuitry in patients. The cellular and molecular basis of selective neuronal vulnerability is beginning to emerge. Yet, there are no effective cures for ALS, which affects more than 200,000 people worldwide each year. Recent studies highlight the importance of the glymphatic system and its proper function for the clearance of the cerebral spinal fluid, which is achieved mostly during the sleep period. Therefore, a potential link between problems with sleep and neurodegenerative diseases has been postulated. This paper discusses the present understanding of this potential correlation.

## 1. Introduction

Amyotrophic lateral sclerosis (ALS) is a complex and a rare neurodegenerative disease, affecting a heterogeneous patient population and developing with many different underlying causes [1,2]. Patients share many common clinical manifestations, as described by the loss of both upper and lower motor neurons and limitations in the motor neuron circuitry. Most patients sporadically develop the disease without a genetic linkage, suggesting the presence of some common underlying problems that are broadly shared. For example, exposure to toxins prior to ALS diagnosis was considered as one of the potential contributors [3]. Likewise, previous injury to the brain, concussions, and traumatic brain injury were also suggested as potential early underpinnings of the progressive neurodegeneration that follows ALS diagnosis [4,5]. Recent evidence has highlighted problems with sleep and a dysfunctional glymphatic system as potential contributors to disease pathology.

## 2. The Importance of Sleep

Sleep is a complex phenomenon. Problems with sleep are estimated to occur in about 17–25% of the population in the United States [6]. In obstructive sleep apnea, which is one of the most common and severe forms of sleep problems, the upper airways collapse during sleep and the rhythm-generating neurons in the brainstem cannot send proper signals to the muscles that are important for the initiation and the rhythmic maintenance of beathing patterns. This drives the brain and the body towards a dangerous hypoxic state. This intermittent hypoxia is one of the underlying causes of major health problems involving cardiovascular, metabolomic, and neuronal dysfunctions.

The “sleep and awake” cycle is precisely orchestrated, mostly by dopaminergic and non-dopaminergic neuromodulation, such as serotoninergic and noradrenergic neuromodulation. Several neurotransmitters are involved in establishing and maintaining this control. The main neurotransmitters are GABA for inhibitory neurons and circuitries and glutamate for excitatory neurons and circuitries. The thalamocortical rhythms important to sleep are regulated by GABAergic neurons [7]. Interestingly, the brain regions that are important for the initiation and regulation of sleep are also the regions that are associated with numerous neurodegenerative diseases. For example, the Lewy body pathology is detected in the midbrain, medulla oblongata, basal forebrain, and neocortex of patients, which are the key regions for the proper control of the “sleep and awake” switch [7].

Glutamate is important as an excitatory neurotransmitter, but when it is not properly cleared from the synaptic cleft, it leads to hyperexcitation and can act as a neurotoxin. Glutamate disturbance is also detected in the presence of sleep apnea [8,9], and the concentrations of glutamate in the plasma of ALS patients have been associated with obstructive sleep apnea [10]. These are interesting findings lay the foundation for a potential link between glutamate toxicity in patients and sleep defects. In fact, riluzole, the first FDA approved drug for ALS, acts on the glutamate clearance system and reduces glutamate toxicity to treat ALS [11].

Different from glutamate, the association of GABA with sleep apnea has also shown a genetic component in that the Phe658Phe polymorphisms in the GABA_B_ receptor 1 gene (*GABABR1*) have been associated with the occurrence of obstructive sleep apnea in patients [12]. Therefore, the association between sleep problems and the dysregulation of the neurotransmitter system is of great interest.

## 3. The Importance of the Glymphatic System

The glymphatic system, -the lymphatic system of the central nervous system-, is comprised of a network of perivascular channels through which CSF mixes with the interstitial fluid (ISF) and drains the metabolic waste out of the brain via meningeal and lymphatic vessels, enabling clearance and refreshment of the brain. The meninges, which cover the cerebral cortex, have lymphatic drainage even though the brain itself is devoid of it [13]. CSF is secreted by choroid plexus, and travels from ventricles of the brain to the subarachnoid space. It enters brain parenchyma, mixes with ISF and flows into the perivenous spaces, exiting via the meningeal lymphatic vessels. This dynamic flow of CSF and ISF clears the metabolites, impurities, aggregates, and cellular toxins [14]. Interestingly, this clearance system is most effective during sleep [7,13].

Since our brains are sustained in CSF, cellular byproducts secreted by neurons and non-neuronal cells of the brain accumulate in CSF, requiring timely and proper clearance. When this process is disrupted, harmful solutes as well as proteins that eventually lead to larger aggregates and protein accumulations are not properly cleared, the pH of the solution shifts to more a non-neutral value, and the brain is put under a stressful condition [7,13,14].

Protein accumulation is emerging as one of the most common problems in neurodegeneration. Even though different proteins accumulate in different diseases, the problem of protein accumulation is shared among many different diseases. For example, there is amyloid-beta (Aβ) accumulation in the brains of Alzheimer’s disease (AD) patients, and tau as well as α-synuclein accumulations are detected in the brains of Parkinson’s disease (PD) patients [15]. TDP-43 pathology, which is initially defined by the accumulation of phosphorylated TDP-43 protein in the cytoplasm of vulnerable neurons, is broadly observed in the brains of patients diagnosed with ALS, FTD (frontotemporal dementia), ALS/FTD, AD, and many others [16,17,18,19,20,21]. Interestingly, TDP-43 pathology is mostly detected in patients who sporadically develop the diseases without a known mutation, suggesting that the protein accumulation problem cannot only be explained by a gene mutation, but, rather, is the result of a more systemic problem [17,18,19,20,21]. Recent evidence begins to suggest that defects with the glymphatic system may be the common culprit and that the inability to clean and clear the unwanted and the toxic content of the CSF may contribute to the protein aggregations detected in the brains of patients. How problems with the glymphatic system relates to ALS pathology has been described in a recent review, which explains the importance and the relevance of glymphatic system for maintaining neuronal health and function [22].

Neurons work with high efficiency, constantly producing metabolic and cellular waste which is secreted via different mechanisms. The amount of waste generated by the neurons and non-neuronal cells of the brain, is reported to be about 7 grams per day [23]. This waste becomes present in the CSF, which is produced and secreted mostly by choroid plexi and other extra-choroidal sites and is a clear, colorless liquid occupying the ventricular system, the subarachnoid space, and the perivascular spaces in the CNS. It is suggested that about 650 ml of CSF is produced per day, mostly after midnight [24]. Since the total volume of CSF remains constant, about 650 ml of CSF must be cleared through the glymphatic system each day, which is an arduous task. CSF is absorbed via the subarachnoid space through the lymph system, the meningeal lymphatics embedded in the dura matter in the brain, and in the spinal arachnoid granulations and spinal meningeal lymphatics in the spinal cord.

The dynamics of CSF flow are tightly regulated and are modulated by motions generated by cardiac-driven forces and the respiratory forces. Therefore, our heartbeat and breathing pattern have a direct impact on CSF flow and clearance [7,13,23]. Interestingly, most of CSF clearance occurs while we are asleep, at the time when the heartbeat and breathing patterns are most synchronized [23]. During sleep, the interstitial space is enlarged by 60% and the flow of ISF and CSF is significantly increased, as shown by two-photon imaging in mice [25]. Similarly, studies in humans have also revealed larger CSF volumes during sleep [26], and that studies with 25 independent subjects reported a faster CSF clearance rate during sleep rather than in an awake state [27].

There was effective amyloid beta (Aβ) clearance from the brain of sleeping mice twice more rapidly when compared to awake mice [25], and sleep-deprived mice displayed significant defects in their glymphatic function [28]. In addition to Aβ, other proteins associated with neurodegeneration, such as α-synuclein, and apolipoprotein E, also displayed reduced clearance after sleep deprivation [29,30]. Even in healthy adults, there was accumulation of the protein tau in their plasma after being acutely sleep deprived [31]. When a single-night of sleep was missed, problems with mood, memory, and attention was detected immediately the next day [32]. These findings, both in mouse models and in humans, postulate a key significance and a strong correlation between the quality of sleep, glymphatic network function, and the clearance of proteins that are prone to aggregate [33].

## 4. Sleep Disturbances and the Glymphatic System

Disturbance in the sleep patterns, such as problems with falling asleep, reduced quality and length, and reduced REM (rapid eye movement) are observed in patients with sleep problems, including patients with obstructive sleep apnea. Sleep problems escalate with age due to the increased stiffness of the cerebral arteries and the reduced amplitude of the cardiac pulse; a significant reduction in glymphatic function is observed [34,35,36]. Therefore, an association between age, sleep problems and reduced effectiveness in the glymphatic system is postulated for many neurodegenerative diseases.

## 5. Sleep Apnea and Neurodegeneration in PD and AD

About 60% of Parkinson’s disease (PD) patients are reported to have sleep disorders [37] with increasing severity as the disease progresses [38]. In one large population-based study, which included 91,273 patients the association between sleep disorders and the development of PD was assessed for over 10 years. This massive study suggested that insomnia lasting for more than 3 months could indeed be a risk factor for developing PD [39]. A more recent study with 328 patients revealed that poor sleep quality was associated with the more severe PD subtype, and there was higher cognitive decline, apathy, and fatigue in patients [40]. Likewise, sleep apnea was observed with a higher percentage among PD patients, when compared to the general population by 2–14% [41]. On the other hand, other studies have suggested that having PD had exacerbated sleep problems because the dystonia of the airways, which develops due to the disease, had an impact on the severity of sleep apnea. In addition, PD is associated with autonomic dysfunction, which impacts the breathing patterns that become dysfunctional in sleep apnea [38,39,40]. Therefore, the relationship between PD and sleep problems does not appear to be single directional. It is possible that once the vicious cycle is established, it turns into a feed forward loop making both the sleep problem and PD pathology much worse over time.

One of the major and first associations with sleep apnea has been the cognitive decline in patients. Significant memory loss and problems with executive function have been observed, which are supported by neuroanatomical changes, such as gray matter loss, a reduction in the size of hippocampus, and the frontal, parietal, and temporal cortices, areas that are associated with AD and FTD [30,31,32]. When rats were exposed to extensive cycles of intermittent hypoxia, as in the case of patients with sleep apnea, there was neuronal apoptosis in the cortical and hippocampal neurons, which was correlated with problems with cognitive and spatial learning [42]. This study generated much interest in the field for connecting the hypoxia that occurs due to sleep apnea with AD. Other studies using different animal models followed with similar results [43], suggesting that the results were not restricted to one species, but, rather, that the correlation between brain hypoxia and neurodegeneration was very significant. Interestingly, about half of AD patients were reported to experience sleep apnea [44]. When sleep apnea was treated in a small subset of AD patients, their cognitive scores improved [45]. Studies with rats and mice revealed that when these animal models were exposed to chronic intermittent hypoxia, there was β-amyloid accumulation detected especially in their hippocampus [46,47], as well as tau hyperphosphorylation [48,49]. Moreover, like patients with severe sleep apnea displaying hippocampal atrophy, rats with disturbed sleep patterns showed a reduction in neuronal arborization, as well as a reduction in the volume of the hippocampus and brainstem respiratory nuclei, together with other brain regions that are linked to cognitive function [50].

Studies emerging from both human studies and animal models revealed that Aβ clearance occurs mostly during sleep and the disruption of sleep leads to increased levels of Aβ in the brain [51]. In line with previous findings, when 3XTg AD mice were exposed to intermittent hypoxia, as in the case of sleep apnea, there was increased Aβ amyloid production [47]. These are very interesting findings, laying a framework for our understanding of the association between sleep apnea and protein accumulation in the brains of AD patients.

## 6. Sleep Apnea and ALS

Sleep problems are also closely linked to ALS [52] and ALS/FTD [53]. The muscles that are important for respiratory function display progressive weakening and degeneration, leading to numerous problems with breathing. Noninvasive ventilation improves patient outcomes. Sleep apnea was found to be more common in ALS patients when compared to the general population [54]. There were significant reductions in total sleep time, oxygen saturation and overall sleep efficiency in ALS patients [55] and interestingly, the mean survival rate in ALS patients with obstructive sleep apnea was significantly shorter than patients without it, suggesting its contribution to the rate of disease progression [56]. A causal association study between obstructive sleep apnea and ALS, including GWAS (genome-wide association study) data from 16,761 ALS patients and 201,194 healthy controls, revealed a link between ALS and sleep apnea, in that having obstructive sleep apnea increases the risk of developing ALS [57].

TDP-43 pathology is one of the most common proteinopathies in ALS, and studies have shown increased levels of TDP-43 protein in the CSF of ALS patients [53,58], suggesting that it has not been properly cleared, and may indeed be used as a prognostic biomarker [58]. Indeed, the proteins that are present in the CSF have been actively studied for the identification of biomarkers and numerous proteins have been found to be differentially present in the diseased CSF [59].

ALS is a complex disease, and the underlying causes of the disease may differ from patient to patient, although some common themes emerge [1,2]. For example, protein accumulation, glutamate excitotoxicity and neuroinflammation are some of the common causes of the disease [1,2,18,21], which are now strongly linked to defects with the glymphatic system-mediated CSF clearance that occurs mostly during sleep [22].

## 7. Early Spine Loss in Diseased Neurons of ALS

In addition to these previously reported common pathologies, there is one more cellular problem that occurs very early in the diseased motor neurons of ALS, and that is detected both in the familial and sporadic cases: the loss of dendritic spines.

One of the first signs of neuronal vulnerability is the loss of spines. Spines are the sites of neuronal connection and are found to be very dynamic even in the adult brain [60,61,62,63,64]. Spines constantly change their morphology and structure, and they undergo repair and pruning, which is more evident in the developing brain, but it is present throughout life [62,64]. Spines and their modulation are exceptionally important for the development and maturation of the motor neurons and the motor neuron circuitry [65]. Numerous studies emerging from the Bellingham laboratory elegantly reveal the dynamic nature of spines and how they are affected early in the disease, especially within the context of ALS with TDP-43 pathology [66,67,68]. Studies utilizing post-mortem human samples isolated from the middle frontal gyrus of FTD patients with TDP-43 pathology revealed significant reduction in the levels of synaptophysin, a presynaptic protein by Western blot analyses. However, there was an increase in the levels of microglia that harbored postsynaptic density protein 95 (PSD95), suggesting an upregulation in the microglial clearance of spines in FTD cases, and an abnormal synaptic pruning when compared to the controls [69].

This spine pruning has been associated with the signals that become present in the spines that needs to be removed [70,71], and the microglial phagocytosis is required to recognize and eliminate the spine [72]. Therefore, spine maintenance has a neuroimmune component [70,71,72].

The cellular and molecular mechanism of this very precise spine clearance has led to the discovery of the “eat me” and “do not eat me” signals, which determine whether spines will be phagocytosed or not [71,72]. For example, the complement proteins C1q and C3 become localized to the tips of the spines that will be recognized and cleared by the microglia, which express the microglia-specific C3 receptor. Similar to the C3 receptor, the TREM2+ microglia also engulf synaptic proteins and the spines. Interestingly mutations in the *TREM2* gene are considered to be a risk factor for ALS [73] and ALS/FTD [74]. The topic of “eat me signals” and how they manifest clearance has been extensively reviewed and is now accepted one of the best orchestrated events to maintain synaptic structures, integrity and function [71,75,76,77].

## 8. The Importance of Sleep for Spine Health

Remodeling of spines to maintaining their health and function represents is an important quest. Spines come in many different forms and shapes and with many different functions. The spines that mediate excitatory and inhibitory synapses are structurally different and their location within the membrane of the soma, axon and apical dendrite may also vary. Therefore, the pruning and remodeling of spines are more complicated than expected. However, this constantly occurs in a healthy brain. Recent studies have begun to show the importance of sleep during this process. For example, two-photon live imaging revealed that dendritic spines were removed at a much faster rate during sleep in adolescent mice [78]. Spines bear the pressure of maintaining connections between neurons and they need to be mostly active while we are awake, but during sleep their strength is reduced, and, in particular, the strength of the spines located in excitatory neurons decrease. This is a required process to maintain their health and overall stability. When the awake cycle is extended, synaptic potentiation becomes difficult to maintain. In addition, during sleep, the endocytosis of excitatory receptors occurs, facilitating structural changes within the spine. However, disrupted sleep patterns and problems with sleep would not allow proper synaptic renormalization, and this, in turn, would add stress to the neurons, as their spines have not been properly cleared and remodeled [79]. Thus, sleep emerges as an important period for spine pruning and remodeling, which is ultimately important for the stability of neuronal connections and the health of the nervous system.

Recent studies with mouse models have shown that sleep deprivation has a significant impact on the density of dendritic spines in the hippocampus [78,79,80]. Similar defects could be present in other regions of the brain, but it is mostly the hippocampus that has been investigated in these studies [80]. In one study, 72 h of sleep deprivation resulted in a reduction of microglia-mediated spine elimination, leading to excessive but unhealthy spines and abnormal neuronal activity in the hippocampus [81]. In both adolescent and adult mice, the short-term memory was impaired, and there were increased excitatory synapses in the granule cells of the dentate gyrus in adolescent mice. One of the most detailed studies on the types of spines that are primarily affected due to a lack of sleep was a study that induced only 5 hours of sleep deprivation in mice and found a significant decrease in select types of spines and in distinct regions of the hippocampus [82], suggesting that even acute sleep deprivation has an impact on individual spine types, and local effects on the structural plasticity of neuronal connections. These studies were eye-opening in the field, revealing the importance and the impact of sleep deprivation, albeit when it is short-term [82]. In these studies, the focus had been on the hippocampal region of the brain, so it is unclear whether other regions of the brain, such as the frontal cortex or the motor cortex display similar problems.

The cerebellum is very plastic, with Purkinje cells which are adorned extensively with excitatory parallel fiber synapses, such that a single Purkinje neuron receives about a 175,000 parallel fiber input [83]. Therefore, it is of great interest to understand whether sleep also has an impact in the cerebellum and the remodeling of the Purkinje cell and their connections. A recent study explored this topic via a 3D reconstructive analysis of more than 7000 spines in the posterior vermis of mice whose sleep was perturbed. Very elegant and detailed studies, both at the level of electrophysiological recordings and the resolution of images, revealed that sleep deprivation also has an impact on the stability and the integrity of spines in the cerebellum, such that during the awake cycle, parallel fibers fire and form connections and sleep is required to prune the synapses that are formed because of coincidence and are destined to be removed [84]. Failure in their proper and timely removal may have an impact on the overall connectivity and neuronal activity, affecting balance and motor neuron circuitry behavior.

## 9. Proposed Relation of Sleep and Spine Pruning in ALS

One of the most interesting studies on the impact of sleep on the motor cortex was conducted on a 36-year-old man with tetraplegia after a spinal cord injury [85]. In this study, 96-channel intracortical microelectrode arrays were placed in his left precentral gyrus, the brain activity was recorded while he learned how to play a new game that required visuo-motor skills, and during his sleep the same day. The study showed that there was a “replay” of the activity in the motor cortex related to the recent visuo-motor skill acquisition and that the related motor cortex areas remained active even during a good sleep [85]. This study was important in demonstrating how dynamic the neuronal activity in the motor cortex is even during sleep and that sleep plays an important role in reinforcing learning behavior for motor function. The dynamics of spine clearance or spine modification could not be investigated in this study, but it revealed the importance of a healthy sleep period for reinforcing the connections formed.

The neurons that become vulnerable to degeneration in the motor cortex lose the integrity and the stability of their spines. Neuronal connections depend on the integrity and the health of spines. Neuronal connections depend on the integrity and the health of spines. Especially in the upper motor neurons (UMNs) that degenerate in ALS patients, the spine loss in their apical dendrites, where most neuronal connections are made, was detected as early as P15 (postnatal day 15) in well-characterized mouse models of ALS [66,67]. Apical dendrite degeneration, followed by spine loss, was observed in the UMNs of both the familial and sporadic cases, and this cellular problem was closely recapitulated in the UMNs of numerous mouse models that are developed based on mutations and pathologies detected in patients [86], suggesting that this is observed not only in a limited subtype of patients, but more broadly in sporadic patients who develop the disease due to very many different underlying causes. These findings are important for two reasons: first, they show that spine loss, especially in the neurons that degenerate in ALS, is an early event, which occurs prior to symptom onset, and second, they show that, regardless of species, similar neuronal problems are observed in the neurons that become diseased and that progressively degenerate. Recent studies have also shown a direct link between sleep deprivation and spine loss, especially a reduction in the density of thin, mushroom, and filopodia-like spine density in the CA1 region of the hippocampus [82].

The suggestion that problems with sleep may indeed contribute to disease pathology in ALS would not be a far-fetched idea. However, one needs to be careful when stating cause or correlation, especially there is an intricate balance affecting both. For example, as the disease progresses, patients develop weakening of the diaphragm muscles and that they may require ventilation for proper breathing. Sleep problems may be exacerbated with disease progression, and increased sleep problems may facilitate disease pathology, generating a vicious feed-forward loop, potentially expediting the rate of disease progression in patients. In fact, it has been shown that patients who develop disease pathology at a much faster rate have problems with sleep in a significantly higher ratio when compared to patients who develop the disease with a much slower pace [52,55,57]. Taking care of ventilation problems in ALS patients early in the disease increases the quality of life and is suggested to increase survival rates in patients.

## 10. Discussion and Conclusions

Sleep now emerges as the prime time for the brain to replenish, repair and prepare for the next day. Problems with sleep lead to the accumulation of many problems, and one of which is the brain’s lack of ability to maintain a healthy balance of spine integrity, health, and stability. When the brain is not able to remodel, remove, and enforce select spines in distinct regions of the brain, this leads to severe problems in hyperexcitability, hypoexcitability and an overall problem with neuronal connectivity. Therefore, when patients lose a good night’s sleep, or, worse, if they have sleep apnea and are subjected to prolonged sleep problems, it becomes harder for the brain to maintain homeostasis. In such cases, they are exposed to severe hypoxic conditions, their CSF cannot be properly cleaned, and problems with neuronal connectivity begin to occur. Thus, it is no surprise that problems with sleep have been linked to neurodegeneration.

Evidence obtained both from different disease models and from studies with patients lays a strong foundation for an important link between sleep problems and ALS. It is possible that problems with sleep, when combined with other underlying causes, may tip the balance towards the cascade of events that leads to neurodegeneration. In such cases, it is possible that the disease could have developed even in the absence of sleep problems, but potentially at a later point. Therefore, it is not possible to propose that sleep problems are the “cause” of the disease, even though they may contribute to the pathology. Once the disease-causing cascade of events is initiated, it also further exacerbates sleep problems due to the weakening of muscles, altering sleep patterns, and making the blood vessels rigid or leaky, which in turn further enhances disease-related pathologies and clinical outcomes. This may lead to the argument that sleep problems become significant because of disease progression. Therefore, there is no one simple explanation for the cause and the correlation of sleep apnea and ALS, but the strong association between the two requires further investigations among ALS patient populations.

Discoveries that emerge both using different model systems and data obtained from healthy controls and patients unequivocally establish the importance of sleep for maintaining the health and the stability of the nervous system, especially the cerebral cortex. Problems with sleep lead to many different problems, which are characterized as the culprit for the development of many neurodegenerative diseases. Since these discoveries are rather recent, we do not have strong evidence to declare whether the relationship is causal or correlative.

Currently, ALS patients are not routinely monitored for their sleep problems at the time of diagnosis or questioned on their sleep problems prior to diagnosis. Most epidemiology studies do not have the sleep component investigated together with other parameters that question genetic and environmental factors. We suggest a more through and more detailed studies are required to assess the impact of sleep problems on the initiation and the rate of disease progression in ALS. This may help in clinical trial design and in the development of inclusion and exclusion criteria and may even be utilized as a potential pharmacokinetic biomarker to investigate the timing and the extent, as well as the rate, of disease progression, especially in sporadic cases of ALS affected with sleep problems. The association between sleep problems and ALS pathogenesis may also help in the identification of prognostic, diagnostic, and, most importantly, predictive and pharmacokinetic biomarkers. Detailed proteomic, metabolomic and lipidomic investigations of CSF isolated from ALS patients with and without sleep problems at different stages of their disease, would help reveal the presence of key biomolecules that are linked to, or correlated with, disease pathology. The prognostic biomarkers could be important for early detection and to address whether a patient will develop ALS; the diagnostic biomarker could help reveal the most relevant underlying causes of the disease for, and the pharmacokinetic biomarkers could indicate about the timing and the extent of degeneration.

Developing strategies to avoid apnea and other sleep problems may also be considered as a potential treatment strategy to help the body and the brain to come out of the potential vicious cycle that exacerbates the rate of disease progression. This may be especially important for patients who develop the disease due to problems with hypoxia, neuroinflammation, protein aggregation and defects with neuronal connectivity. Moving forward, we propose to consider problems with sleep as one of the potential disease-related factors in ALS and incorporate sleep studies in ALS clinics and in our efforts to develop effective treatment strategies for patients.

## Data Availability

No new data were created or analyzed in this study. Data sharing is not applicable to this article.

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
