# Peer review of "Sleep Apnea and Amyotrophic Lateral Sclerosis: Cause, Correlation, Any Relation?"

_brainsci, 2024, doi:10.3390/brainsci14100978_

Round 1

Reviewer 1 Report

Comments and Suggestions for Authors

This is a rather well-written narrative review that investigates the relation of ALS with sleep defects, and makes a very relevant contribution to the field of ALS orientate research (both cause and treatment). However, several issues need to be structurally addressed before acceptance:

1). The title need to change to: Sleep apnea and amyotrophic lateral sclerosis: Cause, correlation and relation?

2). Whole sections and the statements therein needs proper citations, not as seems to be a habit of the author by placing citations at the end of a sentence or subsection. No citations are needed within the paragraph, with each statement. A closing sentence of a section/paragraph should be a statement by the author that is deducted from the foregoing review analysis. Many suggestions are given below.

3). The title of Section 1.1 ‘The importance of sleep’ might not be proper. Perhaps: ‘The physiological processes occurring during sleep’.

4). Section 1.2 fourth paragraph seems similar to Section 1.2 first paragraph, and it is strongly suggested to merge the writing for the function of the glympathic system.

5). Title of section 1.4. Sleep Apnea and Neurodegeneration’ should be renamed to ‘Sleep Apnea and Neurodegenerative PD and AD’.

6). Section 1.6 second paragraph, except for ‘mutations in the TREM2 gene are considered to be a risk factor for ALS(73) is very weakly connected to ALS, and it is suggested to make a stronger connections, otherwise this paragraph will need to be shortened. Omit ‘FTD’ or use common neurodegeneration.

7). The first four paragraphs of Section 1.7 deal not with ALS specifically but set the relationship sleep and healthy spine pruning, while the last two paragraphs are proposing a mechanism relating to ALS. It is suggested to be more concise for the first two paragraphs, and add a new section ‘1.8 Proposed mechanism for the relation of sleep, spine pruning and ALS’, where sections 1.7 paragraphs five, six and seven are used. The addition of a graphic is recommended, illustrating vulnerable brain areas and mechanism.

8). Conclusions should be named ‘Discussion and conclusions’.

9). A recent paper – still under review – submitted to MDPI – Life ‘MDPI Life: Sleep disturbances in Amyotrophic Lateral Sclerosis and prognostic impact, a retrospective study’ (3146375) is worth to cite in section 1.7/1.8, if the editors can facilitate to get this to the author of this review.

10). More details on citations and editing suggestions, are given below:

 Abstract

Change to: ‘Amyotrophic lateral sclerosis (ALS)

Expand and explain a bit more ‘glymphatic system and its proper function’ and adapt the last sentences to be more clear on the content of the paper: e.g.: ‘the glymphatic system a network of perivascular channels where cerebrospinal fluid mixes with the interstitial fluid and drains the metabolic waste out of the brain via meningeal and lymphatic vessels, occurring mostly during sleep. A potential link between sleep disorders/respiratory disturbance and neurodegenerative diseases has been postulated. This paper discusses the present understanding of this potential correlation.

Introduction, replace ‘yet’ with ‘both ALS and neurodegenerative diseases’

Introduction, ‘described by the loss of both upper and lower motor neurons and limitations with the motor neuron circuitry.’ Needs citations

Introduction, ‘Most patients sporadically develop the disease without a genetic linkage, suggesting the presence of some common underlying problems that are broadly shared.’ Needs citations

Introduction, ‘Recent evidence now highlight problems with sleep and a dysfunctional glymphatic system as a potential contributor to disease pathology.’ Needs citations

Section 1.1 title ‘The importance of sleep’ might not be proper. Perhaps: ‘The physiological processes occurring during sleep’

Section 1.1, first paragraph needs citation for each of the statements ‘In obstructive sleep apnea, …. Neuronal dysfunctions‘

Section 1.1, second paragraph needs citations ‘The ‘’sleep and awake’’ cycle … excitatory neurons and circuitries’ as wells as ‘Interestingly, the brain regions that are important for the initiation and regula-tion of sleep are also the regions that are associated with numerous neurodegenerative diseases.’

Section 1.1, third paragraph ‘concentrations of glutamate’, needs elaboration, elevated/diminished. Describe the observed association, if any.

Section 1.1, from ‘Different from glutamate, ‘ make a new paragraph.

Section 1.2 first paragraph ‘The glympathic system … refreshment of the brain‘ needs citations, as well as ‘Interestingly, this clearance sys-tem is most effective during sleep’.

Section 1.2 second paragraph needs citations

Section 1.2 third paragraph ‘Protein accumulation emerges as one of the most common problems in neurodegen-eration. Even though different proteins accumulate in different diseases, the problem of protein accumulation is shared among many different diseases.’ Can be more concisely written.

Section 1.2 third paragraph, write ‘frontotemporal dementia (FTD),

Section 1.2 third paragraph, Need of citation for ‘ Interestingly, TDP-43 pathology is mostly detected in patients who sporadically develop the diseases without a known mutation, suggesting that protein accumulation problem cannot only be explained by a gene mutation, but rather is the result of a more systemic problem. Recent evidence begins to suggest that defects with the glymphatic system may be the common culprit and that inability to clean and clear the unwanted and the toxic content of the CSF may contribute to protein aggregations de-tected in the brains of patients.

Section 1.2 third paragraph, and this sentence needs elaboration and explain the relation: ‘How problems with the glymphatic system relates to ALS pathology has been recently described.

Section 1.2 fourth paragraph seems similar to Section 1.2 first paragraph, and it is strongly suggested to merge the writing for the function of the glympathic system.

Section 1.2 fifth paragraph ‘The dynamics of … most synchronized’ needs citations.

Section 1.2 sixth paragraph, rewrite ‘There was effective protein clearance during sleep’ with citation and to make sense, as this cannot be an opening line of a paragraph.

Section 1.3 write ‘ reduced quality and length sleep periods’, while the whole section needs proper citations, not as seems to be a habit of the author by placing citations at the end of a sentence. No citations are needed within the paragraph, with each statement. A closing sentence of a section/paragraph should be a statement by the author that is deducted from the foregoing review analysis.

Title of section 1.4. Sleep Apnea and Neurodegeneration’ should be renamed to ‘Sleep Apnea and Neurodegenerative PD and AD’

Section 1.4 first paragraph write ‘60%’ and ‘2-14%’, and give citations for ‘41 On the other hand, other studies suggested that having PD had exacerbated sleep problems because dystonia of the airways, which develop due to disease, had an impact on the severity of sleep apnea. In addition, PD is associated with autonomic dysfunction, which impacts breathing patterns that become dysfunctional in sleep apnea.

Section 1.4 second paragraph needs citations for ‘One of the major and first associations with sleep apnea has been the cognitive de-cline in patients. Significant memory loss and problems with executive function have been observed, which are supported with neuroanatomical changes, such as gray matter loss, reduction in the size of hippocampus, frontal, parietal and temporal cortices, areas that are associated with AD and FTD. ‘ and further rewriting: ‘other studies’ need to be defined; ‘not restricted to one species’ should name the species. Also, the jump from human to rat study observations is not correctly done, and is misleading. First mention the rat study observation, give the proper citation and then compare to the human study with citation. Thus reposition ‘like patients with severe sleep apnea displaying hippocampal atrophy’ and ‘as in the case of sleep apnea’. The last sentence should be like ‘These findings lay a framework for our understanding of the association between sleep apnea and protein accumulation in the brains of AD patients.’

Section 1.5 first paragraph, rewrite ‘Sleep problems are also closely linked to ALS52 , as the muscles that are important for respiratory function display progressive weakening and degeneration, leading to numerous problems with breathing(52). Noninvasive ventilation, -explain this procedure-, improves patient outcomes (references).’ and omit FTD(53) as FTD belongs in the previous section.

Section 1.5 first paragraph, abbreviation ‘GWAS’ is undefined.

Section 1.5 first paragraph, give citations, or reposition them, for:‘TDP-43 pathology is one of the most common proteinopathies in ALS, and studies have shown increased levels of TDP-43 protein in the CSF of ALS patients,

Section 1.5 end of first paragraph, rewrite ‘in the ALS diseased CSF’.

Section 1.5 second paragraph, give citations for ‘ALS is a complex disease, and the underlying causes of the disease may differ from patient to patient, albeit some common themes emerge.’ The next sentence also needs citations, or could be partly referring to previously written sections (for protein accumulation and glutamate excitotoxicity, but not for neuroinflammation as is has not been dealt with in this review: ‘For example, as previously discussed both protein accumulation (section 1.x) and glutamate excitotoxicity (section 1.x), as well as neuroinflammation can be proposed as some of the common causes of the disease (references). As outlined below/above these are strongly linked to defects with glymphatic system-mediated CSF clearance that occurs mostly during sleep (22). The next sentence, ‘In addition to these previously reported common pathologies, there is one more cellular problem that occurs very early in the diseased motor neurons of ALS, and that is detected both in the familial and sporadic cases: loss of dendritic spines.’ Needs to be repositioned to section 1.6 as the opening sentence.

Section 1.6 first paragraph, give citations for ‘Spines constantly change their morphology and structure, and they undergo repair and pruning, which is more evident in the developing brain, but it is present throughout life.

Section 1.6 first paragraph, give citation for ‘Studies utilizing post-mortem human samples isolated from the middle frontal gyrus of FTD patients with TDP-43 pathology revealed significant reduction in the levels of synaptophysin, a presynaptic protein by Western blot analyses.’, likely (69).

Section 1.6 second paragraph, give citations and rewrite ‘Spine pruning has been associated with the signals that becomes present in the spines that needs to be removed (reference), and microglial phagocytosis is required to recognize and eliminate the old spine (reference).

Section 1.6 second paragraph, citations are needed: ‘The cellular and molecular mechanism of this very precise spine clearance has led to the discovery of the “eat me” and “do not eat me” signals, which determine whether spines will be phagocy-tosed or not. For example, the complement proteins C1q and C3 become localized to the tips of the spines that will be recognized and cleared by the microglia, which express the microglia-specific C3 receptor. Similar to C3 receptor, TREM2+ microglia also engulf syn-aptic proteins and the spines.

Section 1.6 second paragraph, except for ‘mutations in the TREM2 gene are considered to be a risk factor for ALS(73) is very weakly connected to ALS, and it is suggested to make a stronger connections, otherwise this paragraph will need to be shortened. Omit ‘FTD’ or use common neurodegeneration.

Section 1.7 first paragraph, give citations for ‘Remodeling of spines … sleep during this process.’ As well as for ‘Spines bear … changes within the spine.’ And finish with: ‘Thus, sleep emerges as an important period for spine pruning and remodeling, which is ultimately important for the stability of neuronal con-nections and the health of the nervous system.

Section 1.7 second paragraph, give citations – pleural – as the sentence mentions recent studies ‘Recent studies with mouse models showed that sleep deprivation has a significant impact on the density of dendritic spines in the hippocampus.’ As well as for ‘Both adolescent and adult mice the short-term memory was impaired, and there was increased excitatory synapses in the granule cells of the dentate gyrus in adolescent mice. One of the most detailed studies on the types of spines that are primarily affected due to lack of sleep came from a study that induced only 5 hours of sleep deprivation in mice and found significant decrease in select types of spines and in distinct regions of the hippocampus, suggesting that even acute sleep deprivation has an impact on individual spine types, and local effects on structural plasticity of neuronal connections.

Section 1.7 third paragraph, mentions ‘recent studies’ but cites only (84). Please do proper citing for ‘A recent study delved into this quest with 3D reconstructive analyses of more than 7000 spines in the posterior vermis of mice whose sleep was perturbed. Very elegant and de-tailed studies, both at the level of electrophysiological recordings and the resolution of images, revealed that sleep deprivation also has an impact on the stability and the integ-rity of spines in the cerebellum, such that during awake cycle parallel fibers fire and form connections and sleep is required to prune the synapses that are formed because of coin-cidence and are destined to be removed.’

Section 1.7 fourth paragraph, citation (85) needs to be indicated earlier in the paragraph, e.g., after ‘spinal cord injury’.

Section 1.7 fifth paragraph, rewrite ‘postnatal day 15 (P15)’ and ‘detected in ALS patients(86), suggesting’ and ‘patients who develop ALS due’.

Section 1.7 sixth paragraph, in its entirely should be part of the conclusion, as it summarizes the previous sections and based on this proposes a mechanism.

Section 1.7 seventh paragraph, in its entirely should be part of the conclusion, as it critically discusses limitation to the previously posed mechanism.

The first four paragraphs of Section 1.7 deal not with ALS specifically but set the relationship sleep and healthy spine pruning, while the last two paragraphs are proposing a mechanism relating to ALS. It is suggested to be more concise for the first two paragraphs, and add a new section ‘1.8 Proposed mechanism for the relation of sleep, spine pruning and ALS’, where sections 1.7 paragraphs five, six and seven are used.

Conclusions should be named ‘Discussion and conclusion’.

Author Response

I thank the reviewer for pointing out the mistake in the references and giving me a chance to fix it.

I went over all text and fixed the references and the format.  Attached are my detailed response. Thank you.

Reviewer 2 Report

Comments and Suggestions for Authors

1. I suggest the change of the manuscript's title without using the word "Correspondence": they could describe the manuscript, for example, as "Sleep apnea and ALS: Cause, correlation, any relation?". 

2. The Abstract partially describe the ideas that are discussed in detail in the manuscript. However, the Abstract does not summarize the main idea that is related to the fact that the text represents a "review article" which will be presented to the reader. 

3. The manuscript structure should be reviewed, as there are only two parts: Introduction and Conclusion. 

4. Conclusions are too long (4 paragraphs). 

5. The text of the manuscript is presented as a possible narrative review. However, the structure of the content should be described as a possible "Medical Hypothesis Article". As there are no original data brought by population and epidemiological basis, the most proper way to classify the study should be as a medical hypothesis study. This aspect does not rule out the importance of the manuscript, only states to the reader which type of content is presented. 

Author Response

The reviewer asked that I go over the text and remove double-mentioned sections and work on the structure of the review.  I did as the reviewer suggested and improved the text.  Thank you. 

Round 2

Reviewer 1 Report

Comments and Suggestions for Authors

Most of the previous recommend edits have been granted, and with this the paper is acceptable for publication.

Author Response

thanks

Reviewer 2 Report

Comments and Suggestions for Authors

I thank the author for the revised version of the manuscript. 

There is still the use of only two main divisions in the manuscript: Introduction and Conclusions.

It is necessary to make also adjustments to the Abstract, as it does not describe that the text refers to a review manuscript, for example. 

Author Response

please find in attachment
